# Enhanced Efficiency and Stability of Perovskite Solar Cells Through Neodymium-Doped Upconversion Nanoparticles with TiO_2_ Coating

**DOI:** 10.3390/molecules30102166

**Published:** 2025-05-14

**Authors:** Masfer Alkahtani, Bayan Alshehri, Hadeel Alrashood, Latifa Alshehri, Yahya A. Alzahrani, Sultan Alenzi, Ibtisam S. Almalki, Ghazal S. Yafi, Abdulmalik M. Alessa, Faisal S. Alghannam, Abdulaziz Aljuwayr, Nouf K. AL-Saleem, Anwar Alanazi, Masud Almalki

**Affiliations:** 1Future Energy Technologies Institute, King Abdulaziz City for Science and Technology (KACST), P.O. Box 6086, Riyadh 11442, Saudi Arabia; yalzhrani@kacst.gov.sa (Y.A.A.); sultan0064@gmail.com (S.A.); eb.5555m@gmail.com (I.S.A.); aalessa@kacst.gov.sa (A.M.A.); fsalghannam@outlook.com (F.S.A.); jwr881@gmail.com (A.A.); aqalanazi@kacst.edu.sa (A.A.); mhmalki@kacst.gov.sa (M.A.); 2Department of Physics, College of Science and Humanities, Imam Abdulrahman Bin Faisal University, P.O. Box 1982, Jubail 35811, Saudi Arabia; 2210003574@iau.edu.sa (B.A.); 2210003648@iau.edu.sa (H.A.); 2210003580@iau.edu.sa (L.A.); 3Department of Chemistry, King Saud University, P.O. Box 2455, Riyadh 11451, Saudi Arabia; ghazalyafi@gmail.com

**Keywords:** perovskite solar cells (PSCs), upconversion nanoparticles (UCNPs), TiO_2_ coating, near-infrared (NIR) light harvesting, photovoltaic efficiency and stability

## Abstract

This study presents an effective strategy to enhance the efficiency and stability of perovskite solar cells (PSCs) by integrating neodymium-doped upconversion nanoparticles (UCNPs) coated with a TiO_2_ shell into the mesoporous electron transport layer. The incorporation of neodymium (Nd^3+^) as a novel sensitizer shifts the near-infrared (NIR) absorption band away from the water vapor absorption region in the solar spectrum. This modification enables UCNPs to efficiently convert NIR light into ultraviolet (UV) and blue wavelengths, which are readily absorbed by TiO_2_, generating additional charge carriers and improving photovoltaic performance. The optimized PSCs, fabricated by blending 30% UCNPs@TiO_2_ with commercial TiO_2_ paste, achieved a peak power conversion efficiency (PCE) of 21.71%, representing a 20.4% improvement over the control (18.04%). This enhancement included a 0.9% increase in the open-circuit voltage (*V*_oc_), a 6.6% rise in the short-circuit current density (*J*_sc_), and an 11.9% boost in the fill factor (FF). Additionally, the optimized PSCs exhibited remarkable stability, retaining over 90% of their initial PCE after 900 h in humid conditions, compared to only 70% for the control. These improvements result from enhanced light absorption, reduced moisture infiltration, and lower defect-related recombination. This approach provides a promising pathway for developing highly efficient and durable PSCs.

## 1. Introduction

The escalating global energy demand, traditionally met by conventional sources such as fossil fuels, coal, and natural gas, has raised pressing environmental concerns, including air and water pollution, greenhouse gas emissions, and threats to biodiversity [1,2,3]. A substantial shift toward renewable energy sources, such as solar, wind, biomass, and tidal energy, has become imperative to mitigate these challenges. Among these alternatives, solar energy is particularly promising due to its abundance, sustainability, and potential as a clean and efficient substitute for fossil fuels [3,4,5,6].

Solar energy is primarily harnessed through photovoltaic (PV) technology, directly converting sunlight into electricity. Crystalline silicon (c-Si) technology, including monocrystalline (Mono-Si) and multi-crystalline (Multi-Si) silicon, dominates the market, accounting for approximately 95% of global solar installations. Despite the high efficiency of Mono-Si (~26.7%), the energy-intensive production processes of c-Si-based solar cells, including purification, reduction, and crystallization, present environmental and economic challenges [7,8,9,10]. While gallium arsenide (GaAs) solar cells achieve efficiencies exceeding 30%, their high cost limits large-scale to terrestrial applications [11].

Second-generation thin-film solar cells, utilizing materials such as amorphous silicon, copper indium gallium selenide (CIGS), cadmium telluride (CdTe), and GaAs, offer a more economical approach but still face limitations [12,13]. The emergence of third-generation solar cells, including organic photovoltaics, dye-sensitized solar cells (DSSCs), and perovskite solar cells (PSCs), has marked a breakthrough in the field. Among these, PSCs have demonstrated remarkable records, achieving power conversion efficiencies (PCEs) of up to 27% over the past decade [14,15,16]. This success is attributed to their exceptional material properties, such as their strong visible light absorption, long charge carrier diffusion lengths, and cost-effective fabrication methods [15,16,17]. Despite these advancements, PSCs encounter critical challenges that hinder commercialization, including sensitivity to oxygen, moisture, temperature fluctuations, and ultraviolet (UV) light exposure. Additionally, the narrow absorption range of efficient low-bandgap PSCs limits their ability to fully utilize the solar spectrum, resulting in reduced efficiency and fast degradation [14,15,16,17]. Given that conventional solar cells primarily capture energy from the visible light spectrum (about 48% of total solar radiation), the substantial energy present in the near-infrared (NIR) range (~44%) remains largely untapped [18,19].

Upconversion technology offers a promising strategy to enhance PSC efficiency by converting low-energy NIR photons into higher-energy photons, such as UV and visible light, which can be effectively absorbed by the electron transport layer (TiO_2_) [20] and the photo-active perovskite layer in the PSC device [21,22,23,24]. This process generates additional charge carriers, improving the overall photovoltaic performance of PSCs. A previous study [20] demonstrated this approach by incorporating β-NaYF_4_:Yb^3+^,Tm^3+^@TiO_2_ core–shell UCNPs into the mesoporous TiO_2_ layer, resulting in a significant increase in the power conversion efficiency (PCE) from 13.98% to 16.27% (champion device) and from 13.53% to 16.07% for average fabricated devices. This corresponds to a 16.38% improvement in their champion device performance and an 18% improvement in the average performance across multiple tested devices. Although the results showed a consistent and good enhancement, the achieved efficiencies still remained slightly below the best performances reported for standard perovskite solar cells (PSCs) [22,23]. This limited improvement in the fabricated PSC devices can be attributed to two key factors. First, conventional UCNPs, typically doped with ytterbium (Yb^3+^) and thulium (Tm^3+^), primarily absorb NIR light at 980 nm, a wavelength highly susceptible to energy losses due to water vapor absorption [25,26,27]. Second, the lack of precise control over the crystallinity and uniformity of the TiO_2_ shell in previous designs has led to suboptimal charge transport and interfacial properties, thereby restricting the PSC efficiency and overall performance.

To address these limitations, two synergistic strategies are proposed. First, the NIR absorption of UCNPs will be precisely shifted from 980 nm to 808 nm by incorporating neodymium ions (Nd^3+^), which exhibit a strong absorption at 808 nm, a spectral region with minimal water vapor interference. This targeted spectral tuning is designed to enhance NIR light harvesting while simultaneously mitigating photothermal effects, thereby improving the operational stability of PSCs under ambient conditions. Second, a highly uniform, crystalline TiO_2_ shell will be meticulously engineered and coated onto thulium-doped (Tm^3+^) UCNPs. This core–shell structure ensures the efficient integration into the electron transport layer (ETL), where the upconverted ultraviolet and visible emissions from the UCNPs are effectively absorbed by the TiO_2_ shell. The resulting enhancement in the charge carrier generation and transport is expected to significantly boost the overall photovoltaic performance and long-term durability of the device.

While perovskite materials indeed possess the capability to absorb photons in the near-infrared (NIR) region around 800–810 nm, their absorption coefficient in this spectral range is considerably lower compared to their absorption in the visible spectrum [28,29,30]. Consequently, a substantial portion of NIR photons remains insufficiently utilized, limiting the photovoltaic performance potential of perovskite solar cells. To address this, we strategically employ neodymium-doped upconversion nanoparticles (UCNPs) designed to absorb NIR photons specifically at 808 nm, a wavelength with minimal interference from atmospheric water vapor absorption. The rationale for using upconversion here, despite its lower intrinsic conversion efficiency relative to direct absorption, lies in its ability to transform these weakly absorbed NIR photons into more energetically favorable ultraviolet (UV) and blue photons [31,32]. These higher-energy photons are subsequently efficiently absorbed by the TiO_2_ electron transport layer and the perovskite active layer, thereby significantly enhancing the generation of additional charge carriers. Furthermore, to mitigate potential stability concerns associated with UV-induced photocatalytic activity at the TiO_2_–perovskite interface, our nanoparticles feature a carefully engineered core–shell structure. This design helps isolate the UV-excited TiO_2_ from direct interaction with the perovskite layer, thereby substantially reducing the risk of degradation at the interface and contributing positively to device stability.

This paper presents a simple yet efficient solvothermal method for synthesizing neodymium (Nd)-doped UCNPs with a core–shell structure, followed by the continuous deposition of a TiO_2_ layer. The structural and optical properties of the TiO_2_-coated Nd-doped UCNPs were thoroughly characterized. These engineered UCNPs were then systematically integrated into PSC architectures, with their photovoltaic performance rigorously evaluated and benchmarked against control PSCs. By enhancing the NIR absorption and optimizing the ETL design, this research aims to advance PSC efficiency, contributing to developing more efficient and sustainable solar energy technologies.

## 2. Results and Discussion

The proof of concept for this study is illustrated in Figure 1a. It proposes the integration of UCNPs coated with a uniform layer of TiO_2_ into the mesoporous electron transport layer of PSC devices. In this approach, the absorbed NIR light is upconverted into UV and blue light by Tm ions, which is then reabsorbed by the TiO_2_ layer. This process generates additional electrons, ultimately enhancing the overall photovoltaic performance of the PSCs. Experimentally, the UCNPs were synthesized with a core–shell structure using a solvothermal method, as previously reported [22,27,33] and detailed in Section 3. The incorporation of Nd^3+^ into the core–shell structure of the UCNPs introduces a new sensitizer and an additional NIR absorber at the 808 nm band. This specific wavelength minimizes water vapor absorption in the sunlight spectrum, improving light utilization. As shown in Figure 1b, the Nd^3+^ sensitizer is incorporated into a separate shell during synthesis to maintain a sufficient spatial separation from the upconverting ion, Tm^3+^. The energy losses associated with water vapor absorption arise because conventional UCNPs, typically doped with ytterbium (Yb^3+^) and thulium (Tm^3+^), have absorption bands centered at approximately 980 nm, a spectral region strongly absorbed by atmospheric water vapor, with a first window peak around 1000 nm. This overlap significantly diminishes the intensity of NIR photons reaching the UCNPs, thereby reducing the overall photon conversion efficiency and limiting photovoltaic performance improvements. By shifting the absorption band to 808 nm, where atmospheric water vapor absorption is minimal, our (Nd^3+^) doped UCNPs mitigate these energy losses, resulting in improved light utilization and higher device efficiency. This strategic design minimizes detrimental cross-talk interactions, which are known to decrease the upconversion efficiency of UCNPs significantly [25,26]. Moreover, the addition of the shell to the UCNPs plays a critical role in preventing hydroxyl (OH) groups in water from interacting with the upconverting ions. The presence of OH groups and surface defects can resonate with the intermediate states of the rare-earth ions, leading to poor upconversion efficiency. The shell effectively shields the active ions, preserving their efficiency and contributing to the overall enhancement of the device’s performance [27,34].

The morphology and structural properties of the synthesized UCNPs and UCNPs@TiO_2_ nanoparticles were characterized using transmission electron microscopy (TEM) and X-ray diffraction (XRD) techniques. For TEM imaging, a few drops of each sample suspension were carefully deposited onto carbon grids and allowed to dry before analysis. Figure 2a displays the TEM images of the well-dispersed YLiF_4_:Yb,Tm@YLiF_4_:Nd core–shell UCNPs, exhibiting a uniform octahedral shape with an average particle size of approximately 20 nm. The high dispersion quality and uniformity of the synthesized UCNPs are critical for achieving a consistent and homogeneous coating of the TiO_2_ layer. Subsequently, a uniform TiO_2_ layer with a thickness of approximately 7 nm was successfully coated onto the core–shell UCNPs, as illustrated in Figure 2b. The TEM images of the synthesized UCNPs@TiO_2_ nanoparticles align well with previous studies that employed similar particles for biological applications [35]. The even distribution and conformal coverage of the TiO_2_ shell are essential to maintain the optical and electronic properties of the nanoparticles while enhancing their stability and performance in photovoltaic applications.

The successful deposition and crystallinity of the TiO_2_ coating were further validated by an XRD analysis, presented in Figure 2c. The XRD pattern of the synthesized core–shell UCNPs (black curve) revealed well-defined crystalline peaks that corresponded to the standard pattern of the highly crystalline octahedral phase of YLiF_4_ (JCPDS 01-081-2254). After the TiO_2_ coating, the XRD pattern of UCNPs@TiO_2_ (shown in the red curve) maintained the characteristic peaks of the core UCNPs while also displaying new peaks that aligned with the standard crystalline phase of TiO_2_ (JCPDS card no. 21-1272). The retention of the core UCNPs’ crystalline structure, along with the appearance of distinct TiO_2_ peaks, indicates that the TiO_2_ shell is not only uniformly coated but also well crystallized. This high degree of crystallinity is crucial for optimizing the electron transport properties and enhancing the overall performance of the PSC devices.

The optical properties of the synthesized UCNPs and UCNPs@TiO_2_ were investigated using a custom-built confocal scanning microscope. This setup included an 808 nm laser excitation source, a high-resolution microscope objective, and a custom-built spectrometer, as illustrated in Figure 2d. To prepare the samples for optical analysis, a few drops of each sample were deposited onto a quartz substrate and spin-coated to achieve a uniform, thin film suitable for detailed optical study.

The uncoated core–shell UCNPs exhibited a strong upconversion (UC) emission from Tm^3+^ ions, with a prominent blue emission peak at 475 nm under 808 nm excitation at a low laser intensity of 10 W/cm^2^. This emission corresponds to the radiative transition from the excited ^1^G_4_ state to the ^3^H_6_ ground state. Additionally, weaker emission peaks at 600 nm and 650 nm were observed, attributed to less efficient higher-order transitions, as shown in Figure 3a. Upon coating the core–shell UCNPs with a TiO_2_ layer, a significant quenching of the blue emission from Tm^3+^ ions was observed. This reduction is primarily due to the strong absorption of TiO_2_ in the ultraviolet and blue regions of the spectrum, as depicted by the red curve in Figure 3a. Furthermore, although Tm^3+^-doped UCNPs typically exhibit UV emissions in the 360–375 nm range, this band was not detected in our measurements due to spectrometer limitations.

To explain the underlying photophysical process, the upon 808 nm excitation, YLiF_4_:Tm,Yb@YLiF_4_:Nd UCNPs absorb NIR light and emit both upconverted ultraviolet (UV) and blue light. These emissions excite electrons from the valence band (VB) to the conduction band (CB) of the TiO_2_ shell, generating photoinduced electron–hole pairs [35], as illustrated in Figure 3b. This charge carrier generation plays a crucial role in enhancing photocatalytic and photovoltaic applications. For instance, in the context of perovskite solar cells, the excited electrons in the TiO_2_ conduction band can efficiently transfer to the ETL, typically composed of TiO_2_, enhancing the charge collection and improving the overall photovoltaic performance. Simultaneously, the holes left behind in the TiO_2_ valence band may either recombine or participate in interfacial charge transfer processes, potentially influencing carrier dynamics in the perovskite absorber layer.

We measured the optical transmittance of FTO/UCNP@TiO_2_ films at various UCNP concentrations (0%, 10%, 20%, 30%, 40%, 50%, and 60%), as shown in Appendix A. The results indicate a progressive decrease in the visible light transmittance (400–800 nm) with an increasing UCNP content, which is attributed to enhanced scattering and absorption within the ETL. The UCNPs used are doped with Tm^3+^ ions, which emit UV and blue photons under NIR excitation. These emissions align well with the absorption spectrum of TiO_2_, allowing the upconverted photons to be efficiently utilized for charge generation. These findings suggest that the photocurrent enhancement observed is not solely due to light trapping or optical losses, but rather due to actual upconversion-mediated photon harvesting facilitated by the spectral overlap between the UCNP emission and TiO_2_ absorption.

The synthesized UCNPs@TiO_2_ nanoparticles were incorporated into the fabricated PSC devices to systematically assess their photovoltaic performance. The mesoporous layer of the PSCs was modified with UCNPs@TiO_2_ at varying weight ratios (10%, 20%, 30%, 40%, 50%, and 60%) alongside a control device without UCNPs@TiO_2_. The photovoltaic performance parameters, including the PCE, open-circuit voltage (*V*_oc_), short-circuit current density (*J*_sc_), and fill factor (FF), were evaluated through photocurrent density–voltage (*J–V*) measurements under a standard one-sun illumination (AM 1.5 G).

From Table 1 and Figure 4a–c, the control device exhibited a baseline PCE of 18.0%, with a *V*_oc_ of 1.115 V, a *J*_sc_ of 24.15 mA/cm^2^, and an FF of 67%. With the incorporation of UCNPs@TiO_2_ there was a consistent increase in photovoltaic performance up to a 30% loading (target device). The statistical data of all conditions are illustrated in Appendix A. The PCE increased by 20.4% in the target device, achieving a peak efficiency of 21.7% at 30% UCNPs@TiO_2_, as shown in Figure 4b. This enhancement was accompanied by an increase of 0.9% in *V*_oc_, 6.6% in *J*_sc_, and 11.9% in FF, as illustrated in Figure 4c. The data indicate that a moderate UCNPs@TiO_2_ loading significantly improves the light harvesting and charge transfer within the device. This was mainly due to an improved *J*_sc_ and FF, indicating an enhanced light absorption benefiting from the extra carriers generated with upconversion nanoparticles. Beyond the 30% loading, the performance began to decline. At 40% UCNPs@TiO_2_, the PCE dropped to 19.8%, a 9.5% gain relative to the control but a 9.0% decrease from the optimal 30% loading. The 50% and 60% loadings showed even more pronounced reductions in the PCE to 18.4% (+1.8% from control) and 17.0% (−5.8% from control), respectively. The decrease in the *J*_sc_ by 3.7% and 8.3% at 50% and 60% loadings further supports the hypothesis of the increased charge recombination and hindered charge mobility due to nanoparticle aggregation at higher concentrations.

The substantial enhancement observed in the fill factor (FF) and overall photovoltaic performance arises primarily from the improved interfacial energetics, reduced charge recombination, and optimized charge extraction facilitated by the integration of UCNP@TiO_2_ nanoparticles. Specifically, the incorporation of these nanoparticles leads to the effective passivation of defect states at the ETL–perovskite interface. This passivation reduces trap-mediated recombination pathways, significantly diminishing non-radiative losses. Additionally, as confirmed by the ultraviolet photoelectron spectroscopy (UPS) analysis (vide infra) and explained in Appendix A, the presence of UCNP@TiO_2_ nanoparticles optimizes the energy-level alignment at the interface, enhancing the electron extraction efficiency from the perovskite to the ETL. This energetically favorable alignment mitigates the accumulation of charge carriers at the interface, thereby improving the FF significantly. Moreover, the uniform distribution and controlled morphology of the TiO_2_-coated nanoparticles within the mesoporous ETL further promote efficient charge transport, minimizing the series resistance and enhancing the overall device performance. Thus, the observed improvements in electrical characteristics are consistently supported by the comprehensive interface analysis, energy-level characterizations, and optical assessments, substantiating the mechanistic explanation provided.

Furthermore, an additional critical aspect of this study was the utilization of Nd-doped UCNPs to shift the NIR absorption band to 808 nm, strategically minimizing water absorption interference. Traditional UCNPs that absorb at 980 nm are prone to significant energy losses due to the strong absorption of water at this wavelength, leading to up conversion quenching effects for the UCNPs. By shifting the absorption to 808 nm, where water absorption is minimized, the Nd-doped UCNPs enhance the light-harvesting capability of the PSCs, particularly under natural sunlight where NIR wavelengths contribute substantially to the solar spectrum. The spectral shift not only improved the photon conversion efficiency but also contributed to the observed increases in the *J*_sc_ and PCE at optimal UCNP loadings. The results indicate that this strategic shift in absorption properties enhances the utilization of sub-bandgap light, contributing to a better overall device performance.

The optical properties of the control and target devices (with 30% UCNPs@TiO_2_) were thoroughly evaluated by analyzing their photoluminescence (PL) and internal photon conversion efficiency (IPCE) spectra. The PL emissions of perovskite films, both with and without UCNPs@TiO_2_, were examined to assess the influence of the UCNP incorporation on the luminescence behavior. As depicted in Figure 4d, the steady-state PL spectra of the perovskite films exhibited a distinct and prominent luminescence peak, precisely aligning with the absorption edges of the perovskite photoactive layer. The target device, containing 30% UCNPs coated with TiO_2_, demonstrated a significantly higher PL intensity compared to the control device. This enhancement indicates that the optimal incorporation of UCNPs@TiO_2_ effectively mitigates trap-mediated and non-radiative recombination within the perovskite layer. The reduced recombination losses contribute to an increase in the overall luminescence efficiency, highlighting the functional role of UCNPs in boosting the photoactive layer’s optical performance.

In addition to improved luminescence, the target device with 30% UCNPs@TiO_2_ also exhibited a superior IPCE across a broad wavelength range of 300–800 nm, as shown in Figure 5a. The integrated current density shows an improvement in agreement with the J-V data. The improved quantum efficiency spectrum observed in the device incorporating UCNP@TiO_2_ nanoparticles, characterized by enhanced performance in the near-infrared (NIR) spectral region, highlights the significant impact of defect states located near the perovskite band edges on the device’s photovoltaic behavior. These defect states typically act as recombination centers, capturing photogenerated carriers and reducing the effective charge extraction efficiency. The incorporation of UCNP@TiO_2_ nanoparticles mitigates this effect by effectively passivating interface and bulk defects, thus reducing the density of these recombination centers. Consequently, this defect passivation enhances the charge carrier lifetime and mobility, resulting in an increased quantum efficiency across a broader spectral range, notably around the band edges. Thus, the dual NIR absorption bands not only reflect the upconversion activity but also underscore a critical improvement in defect passivation and charge extraction processes within the perovskite solar cells. Furthermore, defects trap charge carriers, promoting non-radiative recombination and reducing both the open-circuit voltage (*V*_oc_) and overall efficiency. Sub-bandgap photons (e.g., 850 nm) primarily interact with these defect states, causing trapped carriers and increased recombination losses, while above-bandgap photons (e.g., 620 nm) generate free charge carriers more effectively, enhancing the photocurrent [36]. The incorporation of TiO_2_-coated UCNPs embedded within the ETL further mitigates these limitations by converting two low-energy photons (e.g., 808 nm) into a higher-energy photon in the visible range, which is efficiently absorbed by the perovskite layer. The TiO_2_ coating enhances the charge extraction and transport by facilitating a better energy alignment between the UCNPs and the ETL, thereby improving carrier mobility and minimizing interfacial recombination. This process reduces the population of low-energy carriers prone to defect trapping, leading to a significant reduction in non-radiative recombination losses and a notable improvement in the device performance through the enhanced generation and extraction of high-energy, mobile carriers [36,37,38]. The observed enhancement in the incident photon-to-current conversion efficiency (IPCE) across the entire spectral range, rather than exclusively at the wavelength around 475 nm, highlights the multifunctional role of UCNPs integrated into the ETL. While the upconversion mechanism indeed contributes by converting NIR photons into UV and blue photons, thus theoretically enhancing the IPCE significantly around 475 nm, experimental results indicate a broader and more uniform improvement. This broader spectral enhancement arises because the integration of UCNP@TiO_2_ nanoparticles also substantially modifies interfacial energetics, passivates surface defects, and improves the overall charge transport efficiency at the ETL–perovskite interface. Consequently, the reduction in defect-mediated recombination and improvement in the carrier extraction efficiency have a pronounced positive effect throughout the absorption spectrum, overshadowing any specific narrowband enhancements exclusively attributable to upconversion at 475 nm. Therefore, the data strongly suggest that the comprehensive improvement in the IPCE is predominantly driven by the combined effects of the optimized charge transport, defect passivation, and effective interfacial engineering, in addition to optical conversion phenomena facilitated by UCNPs.

Ultraviolet photoelectron spectroscopy (UPS) was employed to investigate the electronic structures and energy levels of TiO_2_ and perovskite layers with and without 30% UCNPs incorporation, as shown in Appendix A. The UPS spectra revealed clear shifts in both the secondary electron cut-off (SECO) and valence band (VB) regions upon the integration of UCNP@TiO_2_ nanoparticles. Specifically, the SECO edge shifted towards lower binding energies after the UCNP inclusion, which is indicative of a reduced work function and improved electron extraction capabilities. Additionally, the analysis of the VB edges demonstrated beneficial adjustments to the energy alignment between TiO_2_ and the perovskite layer, which facilitated efficient charge carrier transfer at their interface. The resulting optimized energy-level alignment contributed significantly to the enhanced photovoltaic performance via the improved *V_oc_* observed in devices incorporating 30% UCNP@TiO_2_ nanoparticles.

We performed comprehensive long-term stability assessments of the optimized perovskite solar cells incorporating UCNP@TiO_2_ nanoparticles under ambient conditions, including humidity-controlled and accelerated aging tests, Figure 5b. The enhanced stability observed, retaining over 90% of the initial PCE after 900 h of exposure, can be attributed primarily to the effective passivation of defect states and the interfacial stabilization provided by the UCNP@TiO_2_ nanoparticles. Specifically, the engineered TiO_2_ shell surrounding the upconversion nanoparticles acts as a protective barrier, significantly mitigating moisture ingress and preventing the direct interaction of moisture with sensitive perovskite layers. Moreover, the optimized energy-level alignment and reduced recombination at the ETL–perovskite interface significantly slow degradation processes, such as ion migration and chemical decomposition, which are commonly accelerated in the presence of humidity and UV radiation.

The enhanced stability of the target device is attributed to the incorporation of Nd^3+^-doped UCNPs, which effectively convert near-infrared light into visible light, thereby enhancing the electron transport within the perovskite layer. Additionally, the UCNPs contribute to passivating grain boundaries and reducing defect densities in the perovskite film, which mitigates degradation under the prolonged exposure to ambient conditions. This dual functionality of light conversion and defect passivation underscores the potential of UCNPs to not only boost device efficiency but also prolong operational stability, making them a valuable addition to PSC technology.

## 3. Materials and Methods

### 3.1. Preparation of LiYF_4_:Yb, Tm (20/0.5%) Core UCNPs

In a 50 mL two-neck flask, 10.5 mL of oleic acid and 10.5 mL of 1-octadecene were mixed and prepared. A total of 1.0 mmol of LnCl_3_, which was composed of lanthanide earth elements that consist of 79.5 wt.% of Yttrium (Y), 20.0 wt.% of Ytterbium (Yb), and 0.5 wt.% of Thulium (Tm), was added to the mixture. After that, this mixture was heated for 40 min in heating mantle at 150 °C under argon gas to produce a clear yellow solution. After cooling the solution to 50 °C, 5.0 mL of methanol solution containing 2.5 mmol of LiOH·H_2_O and 10.0 mL of methanol solution containing 4.0 mmol of NH_4_F were gradually introduced into the mixture. The solution was then mixed for 40 min while the temperature was kept at 50 °C. To remove methanol and residual water from the mixture, the solution was then gradually heated to 150 °C and kept for 20 min. The subsequent step involved heating the reaction mixture to 300 °C for 1h under the presence of argon gas. At the end of the process, the mixture was allowed to cool to room temperature. The synthesized LiYF_4_:Yb, Tm core UCNPs were then collected, washed three times with ethanol, and subsequently re-dispersed in 10 mL of chloroform.

### 3.2. Preparation of LiYF_4_:Yb, Tm (20/0.5%) @LiYF_4_:Nd (20%) Core–Shell UCNPs

A total of 10.5 mL of 1-octadecene and 10.5 mL of oleic acid as the reaction solvent were prepared in a 50 mL two-neck flask. Next, 1.0 mmol of LnCl_3_ (80.0 wt.% of Yttrium (Y), 20.0 wt.% of Neodymium (Nd)) was added into the reaction. A clear yellow solution was observed by heating the solution to 150 °C for 40 min in a heating mantle under argon gas. After cooling the mixture to 50 °C, a 2.5 mmol of LiOH·H_2_O with 5.0 mL of methanol, 4.0 mmol of NH_4_F with 10.0 mL of methanol, and 10 mL of the core solution for upconversion nanoparticles was added into the reaction mixture. The solution was then stirred for 40 min and kept at 50 °C. After that, the solution was then gradually heated to 150 °C and left for 20 min to remove methanol and residual water from the mixture. Then, under argon gas flow, the reaction mixture was heated to 300 °C and kept for one hour. Finally, the mixture was cooled down to room temperature and LiYF_4_:Yb, Tm@LiYF_4_:Nd core–shell UCNPs were collected after washing three times in ethanol and stored in 10 mL of chloroform for further use.

### 3.3. Synthesis of LiYF_4_:Yb, Tm@LiYF_4_:Nd@TiO_2_ NPs

TiO_2_-coated core–shell UCNPs were synthesized following prior works reported in [20,35]. The synthesis involved two main steps: surface modification with cetyltrimethyl ammonium bromide (CTAB) and then coating with a TiO_2_ shell.

#### 3.3.1. Surface Modification of LiYF_4_:Yb,Tm@LiYF_4_:Nd Nanocrystals with CTAB

To modify the surface of the LiYF_4_:Yb,Tm@LiYF_4_:Nd core–shell UCNPs, the particles were dispersed in 10 mL of chloroform, forming a LiYF_4_:Yb,Tm@LiYF_4_:Nd/chloroform stock solution at a concentration of 20 mg/mL. Then, 1 mL of the stock solution and 50 mg of CTAB were added to 20 mL of ultrapure water in a flask under vigorous stirring. The mixture was stirred thoroughly to ensure uniform dispersion, after which chloroform was evaporated in a water bath set at 80 °C, resulting in a transparent solution. The modified nanoparticles were then washed twice with ultrapure water to remove excess CTAB, and the final LiYF_4_:Yb,Tm@LiYF_4_:Nd/CTAB NPs were collected and dispersed in 10 mL of isopropanol.

#### 3.3.2. Coating of a TiO_2_ Shell to Form LiYF_4_:Yb,Tm@LiYF_4_:Nd@TiO_2_ Nanoparticles

The TiO_2_ coating was carried out by dispersing 10 mL of the prepared LiYF_4_:Yb,Tm@LiYF_4_:Nd/CTAB isopropanol solution in a flask, followed by the addition of 2.5 mL of water and 0.3 mL of ammonia hydroxide (25%) under vigorous stirring. Subsequently, 10 mL of titanium diisopropoxide bis(acetylacetonate) (75% in isopropanol, Sigma-Aladdin, Kawasaki, Japan) stock solution (0.01 M in isopropanol) was added dropwise into the flask to ensure the uniform formation of a TiO_2_ shell around the core nanoparticles. The mixture was stirred continuously for 12 h at room temperature to allow the complete reaction of the titanium precursor. Finally, the resulting LiYF_4_:Yb,Tm@LiYF_4_:Nd@TiO_2_ nanoparticles were washed twice with ethanol and deionized water to remove unreacted precursors and byproducts, yielding purified TiO_2_-coated UCNPs.

### 3.4. Preparation of Perovskite Solar Cell

The PSC fabrication began with fluorine-doped tin oxide (FTO) glass substrates measuring 1.7 cm × 2.6 cm. To form a patterned electrode structure and prevent short circuits, 0.5 cm of the FTO layer was selectively etched from the top side of the substrates using zinc powder and hydrochloric acid (4 M). The substrates were then subjected to a thorough cleaning protocol in an ultrasonic bath, starting with sonication in a 2% soap solution for 30 min, then rinsing with distilled water for 10 min, ethanol for 15 min, and acetone for 10 min. After cleaning, the substrates were air-dried to remove residual acetone and further treated in a UV–ozone cleaner (Jelight Company, Inc., Irvine, CA, USA) for 20 min to eliminate organic contaminants and enhance surface activation.

### 3.5. Preparation of Compact Layer via Spray Pyrolysis

To prepare the compact TiO_2_ solution, 600 µL of titanium diisopropoxide bis(acetylacetonate), 400 µL of acetylacetone, and 9000 µL of ethanol were mixed in a vial to ensure uniform dispersion. Meanwhile, the FTO substrates were preheated on a hot plate at 450 °C for 30 min, with the anode area covered to prevent damage. A thin compact TiO_2_ layer was then deposited using the spray pyrolysis technique, applying the solution in three spray cycles to achieve uniform coverage. Following the deposition, the substrates were maintained at 450 °C for 30 min to promote adhesion and crystallization of the TiO_2_ layer, ensuring a stable and well-formed compact layer.

### 3.6. Deposition of Mesoporous TiO_2_ and TiO_2_-Coated UCNP via Spin-Coating

Incorporation of LiYF_4_:Yb, Tm@LiYF_4_:Nd@TiO_2_ NPs into the mesoporous TiO_2_ layer for PSC fabrication. LiYF_4_:Yb,Tm@LiYF_4_:Nd@TiO_2_ NPs were integrated into commercial TiO_2_ paste (18NRT, Dyesol, Canberra, Australia) to create a mixed paste for the mesoporous layer in PSCs. To prepare the nanoparticles stock slurry, the LiYF_4_:Yb,Tm@LiYF_4_:Nd@TiO_2_ NPs were first dispersed in ethanol to achieve a concentration of 10 mg/mL. This slurry was then blended with the commercial TiO_2_ paste at varying weight percentages of 0%, 10%, 20%, 30%, 40%, 50%, and 60%, followed by overnight stirring to ensure uniform distribution. The NPs’ doped mesoporous TiO_2_ layers were deposited onto FTO substrates using the spin-coating technique, where 50 µL of the prepared solution was applied and spun at 4000 rpm for 30 s to form a consistent layer. After deposition, the substrates were annealed at 450 °C for 30 min to promote crystallization and adhesion of the mesoporous TiO_2_ layer, ensuring the structural integrity and performance of the solar cell.

### 3.7. Perovskite, Spiro-OMeTAD Layers, and Gold Back Electrode for Samples

The perovskite and spiro-OMeTAD layers were prepared inside a dry glove box to maintain a controlled environment and prevent moisture interference. The perovskite precursor solution was formulated by dissolving 1.6 M PbI_2_, 1.51 M FAI, 0.04 M PbBr_2_, 0.33 M MACl, and 0.04 M MABr in 1 mL of a solvent mixture composed of DMF and DMSO at a 8:1 volume ratio. The solution was heated to 80 °C for 15 min to ensure complete dissolution. For deposition, 50 µL of the precursor solution was applied to the substrate using a two-step spin-coating process: 2000 rpm for 10 s, followed by 6000 rpm for 30 s. During the last 18 s of spinning, 200 µL of anti-solvent chlorobenzene was dropped onto the film to remove residual DMSO and DMF, leading to better film morphology. The films were then annealed at 100 °C for 10 min and then 150 °C for 10 min to facilitate crystallization. The spiro-OMeTAD hole transport layer was prepared by dissolving 102.72 mg of spiro-OMeTAD in 1200 µL of chlorobenzene. To this solution, 21.36 µL of Li-TFSI solution (prepared by dissolving 520 mg of bis(trifluoromethane)sulfonimide lithium salt in 1000 µL of acetonitrile) and 34.52 µL of 4-tert-butylpyridine (4-tBP) were added to enhance conductivity and stability. Then, 50 µL of the prepared solution was spin-coated onto the perovskite layer at 4000 rpm for 30 s, forming a uniform hole transport layer. Finally, an 80 nm thick gold (Au) back electrode was thermally evaporated, serving as the essential contact for efficient charge collection and overall device performance.

## 4. Conclusions

In conclusion, this study demonstrates a highly effective strategy to enhance perovskite solar cell performance by integrating neodymium-doped UCNPs coated with a uniform TiO_2_ layer. By shifting the UCNP absorption band from the traditional 980 nm to 808 nm, we substantially reduced photon energy losses caused by atmospheric water vapor absorption, enhancing the absorption efficiency in the near-infrared spectral region. Comprehensive electronic and optical characterizations, including the UPS, IPCE, and photoluminescence analyses, confirmed significant interface improvements, notably reducing defect-mediated recombination and improving charge extraction dynamics. Consequently, the optimized devices incorporating 30% UCNP@TiO_2_ achieved a peak PCE of 21.71%, representing a notable 20.4% enhancement compared to the control device (18.04%). Furthermore, the device stability dramatically improved, retaining over 90% of the initial efficiency after 900 h of exposure under ambient conditions, compared to only the 70% retention observed in the control device. These combined advancements highlight the practical value of spectral optimization and precise interfacial engineering, providing a clear pathway towards developing more efficient and stable perovskite solar cells.

## Figures and Tables

**Figure 1 molecules-30-02166-f001:**
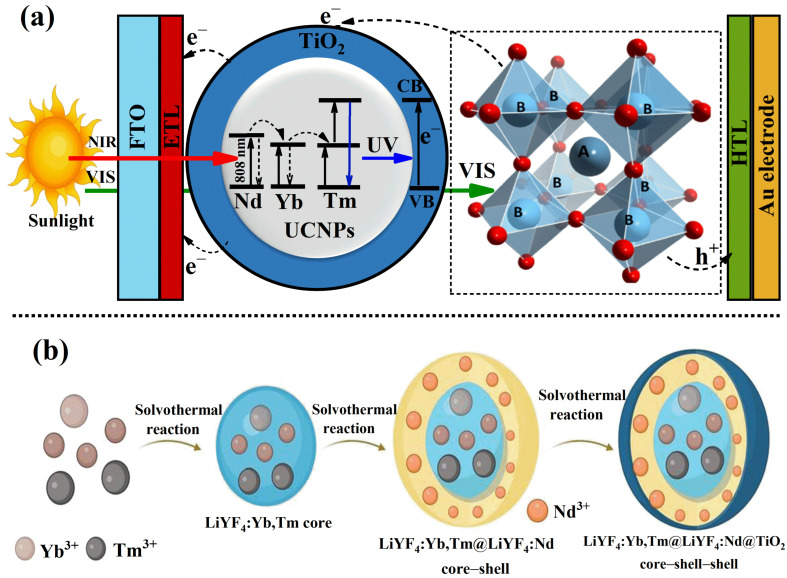
(**a**) A schematic illustration of the proposed proof-of-concept, depicting UCNPs coated with a uniform TiO_2_ layer and integrated into the mesoporous electron transport layer of PSC devices. The introduction of Nd^3+^ as a new sensitizer enables the efficient absorption of NIR light at the 808 nm band, which is then upconverted to blue light by Tm ions. This process generates additional charge carriers through the reabsorption by the TiO_2_ layer, enhancing the photovoltaic performance. (**b**) The synthesis process of core–shell UCNPs via a solvothermal approach. The core–shell structure incorporates Nd^3+^ as a sensitizer to enhance the NIR absorption at 808 nm while preventing interactions between OH groups and upconverting ions, thereby preserving a high upconversion efficiency.

**Figure 2 molecules-30-02166-f002:**
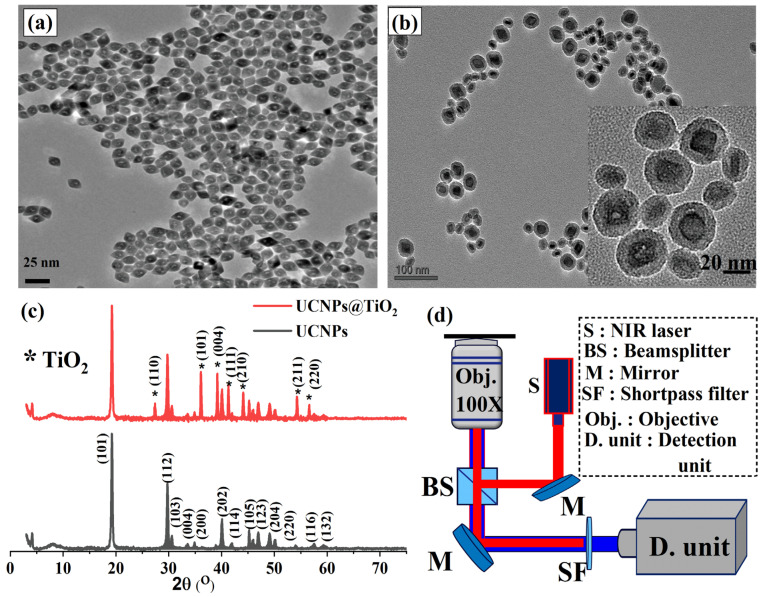
(**a**) A TEM image of well-dispersed YLiF_4_:Yb,Tm@YLiF_4_:Nd UCNPs with a uniform octahedral shape and an average size of approximately 20 nm, demonstrating a high dispersion quality suitable for subsequent TiO_2_ coating. (**b**) A TEM image of UCNPs@TiO_2_, showing a uniform 7 nm TiO_2_ layer coated over the core–shell UCNPs, ensuring a consistent coverage and enhanced stability. (**c**) XRD patterns of the synthesized core–shell UCNPs (black curve) and UCNPs@TiO_2_ (red curve). The diffraction peaks of the UCNPs align with the highly crystalline octahedral phase of YLiF_4_ (JCPDS 16-0334). After coating with TiO_2_, the XRD pattern of UCNPs@TiO_2_ retains the characteristic peaks of the core UCNPs while displaying additional peaks matching the well-crystallized TiO_2_ phase (JCPDS 01-081-2254), confirming a successful and high-quality shell growth. (**d**) A schematic illustration of the confocal scanning microscope used in this study for optical characterization equipped with an NIR laser and a home-made spectrometer.

**Figure 3 molecules-30-02166-f003:**
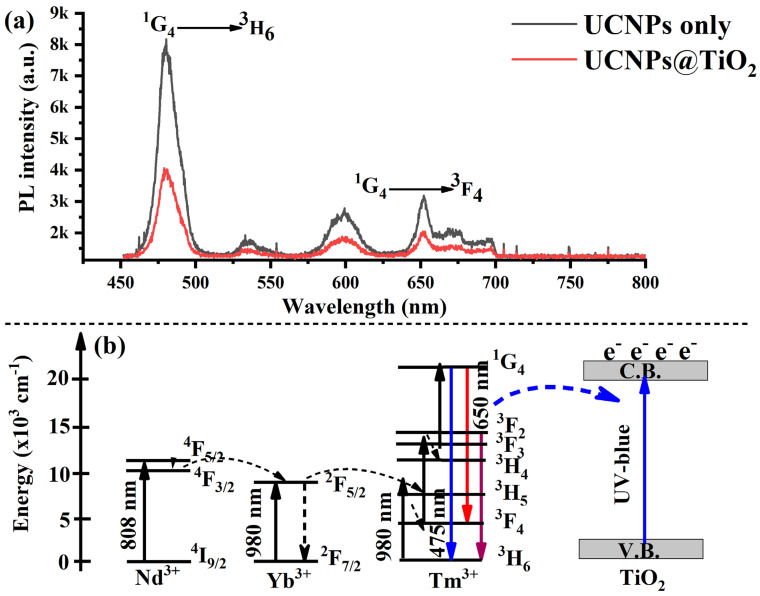
(**a**) Photoluminescence spectra of both uncoated UCNPs and UCNPs@TiO_2_ under an 808 nm laser excitation. The uncoated UCNPs exhibit a strong blue emission at 475 nm, corresponding to the ^1^G_4_ → ^3^H_6_ transition of Tm^3+^ ions, along with weaker transitions at 600 nm and 650 nm. The TiO_2_-coated UCNPs showed a UC emission with a reduced blue intensity due to TiO_2_ absorption (**b**) A schematic illustration of the upconversion process in YLiF_4_:Tm,Yb@YLiF_4_:Nd UCNPs under an 808 nm excitation. The emitted upconverted UV light excites electrons in the TiO_2_ shell, promoting the charge transfer to the conduction band and generating electron–hole pairs. This process enhances charge collection and extends the spectral response of the perovskite solar cell, improving its overall efficiency.

**Figure 4 molecules-30-02166-f004:**
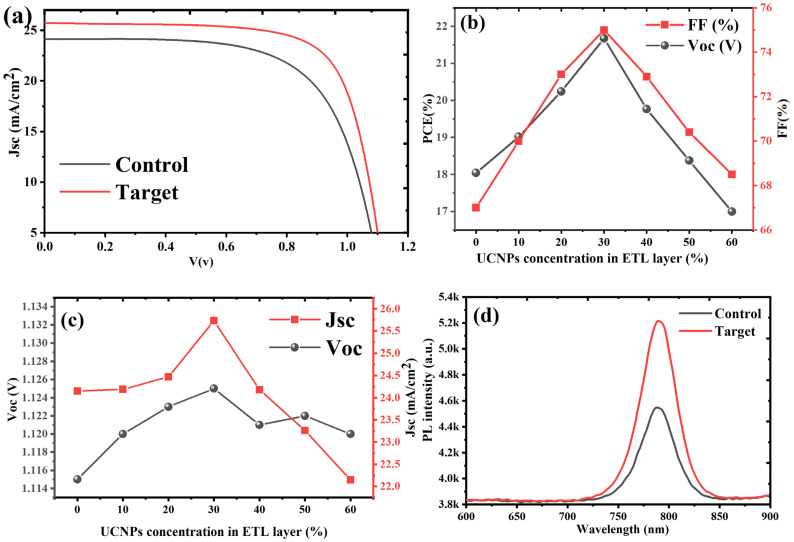
The photovoltaic performance analysis of PSC devices incorporating varying concentrations of UCNPs@TiO_2_ in the ETL. (**a**) J-V curves comparing the control device and the target device with the optimal UCNPs@TiO_2_ concentration, demonstrating an enhanced short-circuit current density (Jsc) and Voc in the target device. (**b**) The relationship between the UCNPs@TiO_2_ concentration in the ETL and the PCE and fill factor, showcasing a peak PCE at 30% loading, with declines at higher concentrations due to nanoparticle aggregation. (**c**) Trends in Jsc and Voc as a function of the UCNPs@TiO_2_ concentration, highlighting the maximal Jsc improvement at 30% loading. (**d**) The steady-state PL graphs of the perovskite exhibited a distinct and prominent luminescence peak, aligning with the absorption edges of the perovskite photoactive layer. The target device with 30% of UCNPs coated with TiO_2_ displayed a higher PL intensity compared to the control device.

**Figure 5 molecules-30-02166-f005:**
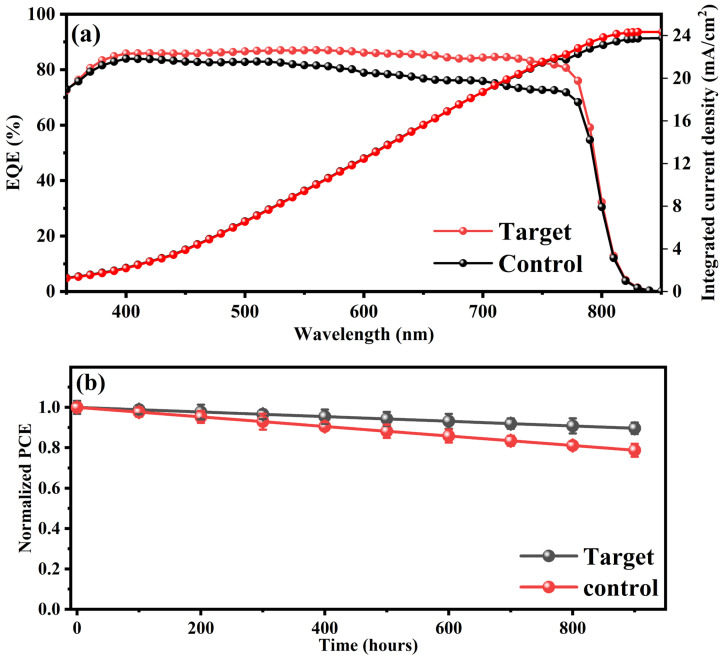
(**a**) The IPCE spectra (left axis) and integrated current density (right axis) of the target device with 30% UCNPs@TiO_2_ and the pristine control device across a broad wavelength range of 300–800 nm. The target device exhibits a superior IPCE performance, particularly within the 350–800 nm region, attributed to the two distinct NIR absorption bands, which indicates that defect states near the band edges in perovskite solar cells play a crucial role in device performance. (**b**) The long-term shelf stability assessment of PSCs under unregulated humidity conditions, comparing the target device with 30% UCNPs@TiO_2_ and the control device over 900 h. The target device demonstrates remarkable stability, retaining over 90% of its initial PCE, while the control device exhibits a significant decline, maintaining only about 70% of its original efficiency.

**Table 1 molecules-30-02166-t001:** Photovoltaic performance metrics of perovskite solar cell (PSC) devices incorporating varying weight ratios of UCNPs@TiO_2_ in the mesoporous layer.

Sample	PCE (%)	*V*_oc_ (V)	*J*_sc_ (mA/cm^2^)	FF (%)
Control device	18.0	1.115	24.15	67.0
Device with 10% UCNPs@TiO_2_	18.9	1.120	24.19	70.0
Device with 20% UCNPs@TiO_2_	20.0	1.123	24.47	73.0
Device with 30% UCNPs@TiO_2_	21.7	1.125	25.73	75.0
Device with 40% UCNPs@TiO_2_	19.8	1.121	24.18	72.9
Device with 50% UCNPs@TiO_2_	18.4	1.122	23.26	70.4
Device with 60% UCNPs@TiO_2_	17.0	1.120	22.15	68.5

## Data Availability

Data are contained within this article.

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
