# Peer review of "Enhanced Efficiency and Stability of Perovskite Solar Cells Through Neodymium-Doped Upconversion Nanoparticles with TiO2 Coating"

_molecules, 2025, doi:10.3390/molecules30102166_

Round 1

Reviewer 1 Report

Comments and Suggestions for Authors

The reported performance lags far behind the current status. The importance is very low for the PSC field. It cannot be published in Molecules.

Author Response

Dear reviewer,

Best regards,

Masfer

Reviewer 2 Report

Comments and Suggestions for Authors

The work is well written and well organized. However, to ensure the reliability of the results, a statistical analysis should be included (one sample for each different composition is not sufficient).

In Figure 5a, add the integrated current density.

Author Response

Dear reviewer,

Best regards,

Masfer

Reviewer 3 Report

Comments and Suggestions for Authors

This manuscript presents an intriguing study on the enhancement of perovskite solar cell (PSC) performance by incorporating upconversion nanoparticles (UCNPs) into the electron transport layer (ETL). The concept of utilizing near-infrared (NIR) light via upconversion is highly meaningful and innovative, especially considering the limited number of reports that demonstrate effective integration of UCNPs into PSC devices. However, I believe several critical issues need to be addressed to validate the claims and improve the scientific rigor of the study. Therefore, I recommend a major revision with the following concerns.

1. The authors attribute the increase in Jsc to NIR absorption via upconversion by UCNPs doped in the ETL. However, UCNPs may also reduce ETL transparency by acting as scattering or absorbing centers. Is the photocurrent enhancement really due to upconversion, or simply from light trapping or optical loss? To clarify this, transmittance spectra of FTO/UCNP@TiOâ‚‚ at various concentrations should be provided, enabling a direct comparison between optical gain and loss.

2. In Fig. 4d, the UCNP-doped sample shows higher PL intensity than the control. This is counterintuitive because efficient electron transfer from perovskite to ETL typically results in lower PL intensity. Then, why does PL increase while Voc also improves? These results appear contradictory. The authors need to explain this discrepancy, possibly by discussing interface passivation or energy level alignment effects that could decouple PL behavior from Voc trends.

3. In Fig. 5a, IPCE improves across the full spectrum, with the largest gain near the absorption edge, not around 475 nm where strong upconversion is expected (Fig. 3a). If upconversion is the main effect, why is the enhancement not wavelength-selective? Author's result suggests that UCNPs may contribute more to charge transfer improvement than optical conversion. Additional discussion and data are needed to clarify why the increase in IPCE at 475 nm is not prominent compared to other regions.

Author Response

Dear reviewer,

Best regards,

Masfer

Reviewer 4 Report

Comments and Suggestions for Authors

This manuscript focuses on the engineering of the electron transport layer of a perovskite solar cell using neodymium-based upconversion nanoparticles for enhanced power conversion efficiency and stability. The topic is interesting in terms of the insight given. However, there are sections of the investigation that need further clarification and the presentation needs major improvements/revisions. Overall, I think that the present work should not be considered for publication in the journal “Molecules”. The following comments should be considered.

  • The authors write that two synergistic strategies are applied for enhanced PCE of the solar cells. (1) shifting of the absorption spectrum of the upconversion particles from 980 to 808 nm, (2) utilization of the emitted UV light by its absorption from TiO2, for enhanced change carries production. For each strategy, the following comments should be addressed. (a) what is the rationale to use the upconversion phenomenon for this specific case of application since 800-810 nm is absorbed by most of the high-performance perovskite active layers. Of course, the upconvertion phenomenon is of much lower efficiency compared to the direct light absorption. (2) It is well known that the excitation of charges from the valence band to the conduction band of TiO2 under UV light (photocatalytic activity) creates highly important issues related to the stability of perovskites (e.g., perovskite decomposition near perovskite/TiO2 interface).
  • In any case, the demonstrated increase in the electrical characteristics of the solar cells using the under-consideration approach seems too high. The interpretations given on page 8 are poor from the scientific evidence point of view. Why did the FF increase so highly? The accompanied characterizations presented in the next pages do not convince. For example, why do trap-mediated and non-radiative recombination within the perovskite active layer mitigate? Please explain better the mechanism.
  • The authors should better explain the sentence, “The enhanced quantum efficiency spectrum observed in the target device, featuring two distinct near-infrared (NIR) 280 absorption bands, indicates that defect states near the band edges in perovskite solar cells 281 play a crucial role in device performance.”.
  • The stability results are not sufficiently discussed and supported. Please revise.
  • The article title is too general and does not represent the content of the present work at an adequate level.
  • Regarding the keywords, the authors should avoid using multiple words (e.g., using and).
  • In the introduction part, the authors write about “limited improvement of PCE” when referring to approaches found in the literature (values on the order of 16.38%), but their approach also results in a corresponding improvement (18%). Please revise.
  • The energy losses due to water vapor absorption should be explained in the text.
  • The conclusion part is poor from the scientific evidence point of view. The authors should not focus on presenting only the variations of the electrical characteristics of the solar cells. Please revise.
  • Abbreviation should be defined once in the abstract and main text, and used thereafter.

Author Response

Dear reviewer,

Best regards,

Masfer

Round 2

Reviewer 3 Report

Comments and Suggestions for Authors

Thank you for providing the revised manuscript.
I have carefully reviewed the revised version, and I can confirm that the authors have appropriately addressed the previously raised concerns. The manuscript is now scientifically sound.

In my opinion, the current form of the manuscript is suitable for publication.

Reviewer 4 Report

Comments and Suggestions for Authors

The authors tried to improve their manuscript and most of my comments have been addressed at an adequate level. Thus, I can now recommend the publication of this work in the journal "Molecules".